# Gene Conversion Explains Elevated Diversity in the Immunity Modulating *APL1* Gene of the Malaria Vector *Anopheles funestus*

**DOI:** 10.3390/genes13061102

**Published:** 2022-06-20

**Authors:** Jack Hearn, Jacob M. Riveron, Helen Irving, Gareth D. Weedall, Charles S. Wondji

**Affiliations:** 1Department of Vector Biology, Liverpool School of Tropical Medicine, Liverpool L3 5QA, UK; jacob.riveron_miranda@syngenta.com (J.M.R.); helen.irving@lstmed.ac.uk (H.I.); charles.wondji@lstmed.ac.uk (C.S.W.); 2LSTM Research Unit, Centre for Research in Infectious Diseases (CRID), Yaoundé P.O. Box 13591, Cameroon; 3School of Biological and Environmental Sciences, Liverpool John Moores University, Byrom Street, Liverpool L3 3AF, UK; g.d.weedall@ljmu.ac.uk

**Keywords:** population genomics, immunogenetics, gene conversion, elevated diversity, parasite–host interactions, mosquito biology, vector biology

## Abstract

Leucine-rich repeat proteins and antimicrobial peptides are the key components of the innate immune response to *Plasmodium* and other microbial pathogens in *Anopheles* mosquitoes. The *APL1* gene of the malaria vector *Anopheles funestus* has exceptional levels of non-synonymous polymorphism across the range of *An. funestus,* with an average π_n_ of 0.027 versus a genome-wide average of 0.002, and π_n_ is consistently high in populations across Africa. Elevated *APL1* diversity was consistent between the independent pooled-template and target-enrichment datasets, however no link between *APL1* diversity and insecticide resistance was observed. Although lacking the diversity of *APL1*, two further mosquito innate-immunity genes of the gambicin anti-microbial peptide family had π_n_/π_s_ ratios greater than one, possibly driven by either positive or balancing selection. The cecropin antimicrobial peptides were expressed much more highly than other anti-microbial peptide genes, a result discordant with current models of anti-microbial peptide activity. The observed *APL1* diversity likely results from gene conversion between paralogues, as evidenced by shared polymorphisms, overlapping read mappings, and recombination events among paralogues. In conclusion, we hypothesize that higher gene expression of *APL1* than its paralogues is correlated with a more open chromatin formation, which enhances gene conversion and elevated diversity at this locus.

## 1. Introduction

In 2018, there were 228 million cases of malaria worldwide, leading to 405,000 deaths. The majority of the cases (93%) and deaths (94%) occurred in Sub-Saharan Africa, and *Plasmodium falciparum* was overwhelmingly responsible (>99%) [1]. *P. falciparum* is vectored by *Anopheles* mosquitoes, and vector competence (susceptibility to *Plasmodium*) varies greatly between species, due to the action of immune genes [2,3]. Furthermore, rising insecticide resistance may increase or decrease the vector competence of *Anopheles* mosquitoes by stimulating or blocking mosquito immune responses [4].

Insects lack the adaptive immune response of vertebrates, relying entirely on a powerful, innate immune system to fight pathogens. Mosquitoes are no exception, employing a complement-like system acting in the hemolymph that identifies and targets pathogens for destruction [5]. In the *Anopheles* species the complement-based response to *Plasmodium* and pathogenic microbes is mediated by a dimer of *Anopheles Plasmodium*-responsive leucine-rich repeat protein 1 (APL1) and Leucine-Rich Immune Molecule 1 (LRIM1), which form a complex with the thioester-containing protein 1 (TEP1) [6], a homologue of the vertebrate complement protein C3. Together, this complex recognizes pathogens of bacterial and fungal origin, in addition to the ookinete stage of *Plasmodium*, and triggers a cascade that eliminates them through either lysis or melanization [7,8]. Another component of the *Anopheles* innate immune systems is the anti-microbial peptides (AMPs), which have a range of antibacterial, antifungal and antiviral activities [9]. They help to make insects extremely resistant to bacterial infection through multiple mechanisms, including membrane destruction, interfering with pathogen metabolism and by targeting cytoplasmic components [9]. Insect AMPs are split into five families, which are variable in presence and copy number across the mosquito species [10]. A cysteine-rich AMP named gambicin was marginally lethal to *Plasmodium berghei* ookinetes [11]. Despite these defense mechanisms, *P. falciparum* is still well able to evade detection by mosquito immunity complexes through the action of the *Pfs47* surface protein and mosquito receptor in a ‘lock and key’ fashion [12,13].

Among the *Anopheles* mosquitoes, the sibling species *Anopheles gambiae* and *Anopheles coluzzii* of the *An. gambiae* species complex have the best studied set of *APL*, *LRIM* and *TEP* genes. Silencing any one of these genes in these species enhances *Plasmodium* infectivity [14,15]. Of particular note are the presence of *Plasmodium*-resistant strains of *An. gambiae sensu lato,* due to the circulating alleles of these genes [16,17]. The *APL1* and *TEP1* genes have elevated genetic diversity compared to background levels and have undergone selective sweeps [18,19]. In the ancestral lineage to the *An. gambiae* complex and *Anopheles christyi,* the *APL1* locus duplicated to form three copies (labelled A, B and C) [20], but remains as a single locus in the other *Anopheles* species, including *Anopheles funestus*. Outside the *An. gambiae* complex, RNAi-mediated silencing of the single-copy APL1 in *Anopheles stephensi* results in increased mortality, which is rescued by antibiotics [20]. Specifically, APL1 in *An. stephensi* modulates the abundance of the *Klebsiella* and *Cedecea* bacteria of the Enterobacteriaceae family [21], both of which are naturally found in *Anopheles* microbiomes. Reyes Ruiz et al. (2019) proposed two models for how the APL1/LRIM1/TEP1 complex acts to eliminate pathogens. In the first model, the complex acts as a recognition molecule that changes conformation on pathogen-binding which activates a zymogen protease; in the second model, the pathogen surface is recognized by another molecule that activates a protease of hemolymph APL1, leading to the release of a processed form of TEP1 in the pathogen’s vicinity [8].

The observations of exceptional polymorphism and maintenance of alleles at the *APL1* and *TEP1* loci in the *An. gambiae* complex were consistent with gene conversion versus alternative possibilities, such as balancing selection [18,19]. However, balancing selection is thought to explain similarly raised levels of genetic diversity seen in the AMPs of four populations of *Drosophila melanogaster* [22] and is possibly acting on gambicin in *An. gambiae* [23]. By contrast to *APL1* and *TEP1* loci, the third component of the complement complex, *LRIM1*, has not been associated with higher diversity in *An. gambiae* [19,24]. An adaptive role for *APL1*- and *TEP1*-elevated diversity may reflect protection against a variety of microbial pathogens that vary across a mosquito species’ ranges and seasonality [18,20]. however, high diversity of *APL1* is not a genus-wide trait; in *An. stephensi* the single-copy of *APL1* exhibits low genetic diversity in an Iranian-sampled population [20]. This could reflect a locally stable microbiome of enteric taxa that places *APL1* under purifying selection. In *An. stephensi*, APL1 has a dual function against *Cedecea* and *Klebsiella* enterobacteria, and *Plasmodium,* which was demonstrated with *Plasmodium yoelii* following the depletion of APL1 [20,21]. *An. gambiae s. l.* APL1 differs as the three APL1 proteins protect against *Plasmodium* infection, but not pathogenic bacteria [20], meaning an alternative mechanism must perform anti-bacterial responses.

*An. funestus* is one of the major vector species of *P. falciparum* in sub-Saharan Africa. It is widespread across Sub-Saharan Africa, and in several regions, it is the dominant vector of malaria [25,26]. As a member of the *An. funestus* group it is phylogenetically distant from *An. gambiae*, *Anopheles arabiensis* and *An. coluzzii* of the *An. gambiae* species complex. Historically, it has been more difficult to breed and maintain laboratory colonies, until the introduction of forced egg-laying techniques [27]. Together, these factors have left *An. funestus* less well-studied in many aspects of its biology, relative to the other malaria vector species. Indeed, a chromosome-level genome assembly became available much later than for the *An. gambiae* complex species [28]. Despite this prior resource deficit, the molecular basis of resistance to pyrethroid insecticides is, arguably, best understood in this species, with several metabolic resistance-conferring variants identified in cytochrome p450 and glutathione-S-transferase (GST) genes using both whole genome sequencing and targeted enrichment sequencing approaches and subsequently functionally validated [29,30,31]. Over-expression of GSTs creates a possible point of crosstalk between resistance and immunity, as the over expressed GSTs may neutralize reactive oxidative species (ROS), which are a key component of insect immune response to *Plasmodium* [4,32]. There may also be an energetic trade-off between the over expression of metabolic resistance genes such as GSTs and the robustness of immune responses to *Plasmodium* and other pathogens [4,32]. *An. funestus* encodes single-copy orthologues of the *APL1*, *LRIM1* and *TEP1* complement genes with paralogues for each gene (VectorBase genome annotation, version 51). This is particularly so for the *TEP*-family genes, which may reflect a Diptera-wide trend for expansion in this family [33,34]. The *An. funestus* genome also encodes 11 AMP genes of the defensin, cecropin, attacin and gambicin families, but lacks the diptericin gene (in annotation version 51) [28].

Here, we assessed the genetic diversity of the *APL1*/*TEP1*/*LRIM1* and AMP innate immunity genes in *An. funestus* for the first time, using a combination of genomic and expression datasets. Among the AMP genes we found that two gambicin genes have elevated π_n_/π_s_ ratios concordant with ongoing selection alongside a *high* expression of cecropins across the *An. funestus* populations. Principally, we show that the key *Plasmodium* and Gram-negative bacteria response gene *APL1* has high non-synonymous (π_n_) and synonymous (π_s_) diversities versus genome-wide averages for this species. Multiple sources of evidence, including read mappings, shared polymorphisms and recombination between the *APL* paralogues, suggest that non-homologous gene conversion plays a key role in the elevated diversities.

## 2. Materials and Methods

### 2.1. Identifying APL, LRIM, TEP and AMP Genes in An. funestus

We selected immune genes that were directly annotated as *APL*, *LRIM*, *TEP* and AMP genes (a) from the *An. funestus* AfunF3.1 gene-set; (b) the genes that were designated as orthologues of *An. gambiae* (AgamP1.10) genes of each category in VectorBase; and (c) the *An. funestus* paralogues for genes selected by criteria (a) and (b). VectorBase defines the orthologues and paralogues using OrthoMCL [35].

### 2.2. Variant Prediction from Genome-Wide Pooled-Sequencing

Read data for pooled-template whole genome sequencing (PoolSeq), target-enriched individual genome sequencing (SureSelect) and RNA-sequencing (RNAseq) analyses are available in the European Nucleotide Archive under accessions PRJEB13485, PRJEB24384, PRJEB35040, PRJEB24351, PRJEB24520, PRJEB47287, PRJEB48958 and PRJEB24506 [29,31,36,37]. The pooled-sequencing data for ten F_0_ populations of *An. funestus* from eight locations and two lab strains [29,36] were aligned to the AfunF3.1 genome assembly [28] with BWA (0.7.17) (sampling locations and alignment metrics, Appendix A). Alignment bam files were sorted, duplicates removed with Picard (2.18.15-0) [38] and converted to mpileup format in Samtools (1.9) [39]. Variants were identified with Varscan (mpileup2cns, version 2.4.3) [40,41] with a *p*-value threshold of 0.05 and a minimum allele frequency of 0.01. Variants were left-aligned, normalized, split into biallelic sites, and SNPs filtered within 20 bp of an indel in bcftools (1.9) [39]. The SNP effects were predicted in SnpEff (v4.3) [42] using a custom *An. funestus* database created from the AfunF3 genome. The population genetic parameters were estimated for annotated genes using SNPGenie (version 2019.10.31) [43], filtered variants, and the AfunF3.1 genome annotation. The additional genes were noted if they had high diversities and could be linked to immune function. Per gene and exon average read coverage depths were calculated in Jvarkit (version d9efbd3, https://github.com/lindenb/jvarkit, accessed on 23 April 2020) “bamstats05.jar” from bam alignment files to compare per-gene coverages versus genome-wide averages.

### 2.3. Targeted Sequencing of Candidate Resistance Genes and Regions

To identify the *APL1* variant sites with allele frequencies significantly associated with resistant phenotypes, we analyzed a targeted enrichment experiment, based on the genes with a potential role in insecticide resistance (see [37] for further details). For the populations of *An. funestus* from Malawi and Uganda, we selected ten mosquitoes that died after 60 min exposure to permethrin and ten mosquitoes still alive after 180 min exposure. Due to lower resistance levels in Cameroon, the mosquitoes that died after 20 min exposure to permethrin or were alive after 60 min were collected. Ten individuals each from the susceptible FANG lab strain originating in Angola, and the FUMOZ pyrethroid-resistant strain originating in Mozambique were also included [44]. In short, a selection of potentially resistance-related genes, including heat shock proteins, odorant binding proteins, detoxification genes and immune response genes, and all of the known target-site resistance genes’ sequences from *An. funestus* [37]. Additionally, all of the genes in the major quantitative trait loci associated with pyrethroid resistance were included. These were a 120 kb region BAC clone of the *rp*1 (resistance to pyrethroid 1) locus containing the *CYP*6 *P*450 gene cluster on chromosome 2R and the 113kb region BAC clone sequence for *rp*2 on chromosome 2L [45]. A total of 1302 target sequences were included. The baits were designed using the SureSelect DNA Advanced Design Wizard of the eArray program of Agilent; the library preparation and sequencing were performed by the Centre for Genomic Research (CGR), University of Liverpool, using the SureSelect target enrichment custom kit. The libraries were pooled in equimolar amounts and sequenced in 2 × 150 bp paired-end fragments on an Illumina MiSeq with 20 samples per run (version 4 chemistry).

The alignment, sorting and duplicate removal of the SureSelect data for 80 individuals were the same as for the pooled sequencing. The variants were called in freebayes (v1.3.2) and filtered for a phred-scaled quality score greater than 20 with ‘vcffilter’ of vcflib (v1.0.0) [46]. Only the genes with an average coverage of over 500 calculated in Jvarkit from all of the pooled individuals were retained for population genetic analyses. The diploid SureSelect variant data were then phased, using WhatsHap [47]. The phased SNP-only genotypes for 160 haploid genomes were converted into fasta sequences in bcftools. The immune genes of interest were extracted from haplotype genomes using GffRead [48] and aligned in muscle [49] to ensure correct positioning across the haplotypes. Kimura’s 2-parameter distance haplotype networks were constructed in the R package pegas, as were per gene Tajima’s D estimates and *p*-values to indicate the population structure and the directional or balancing selection at each locus [50].

### 2.4. F_st_-Based Associations between Resistant and Susceptible Mosquitoes within Each Country

Non-synonymous and synonymous annotated SNPs for immune genes separately, and the whole of chromosome 2, were input to the R package poolfstat (v2.0.0) in variant calling format (VCF) for global F_st_ estimates from African populations [51]. An F_st_-based association study was performed on *APL1* variants between the ten susceptible (dead) and ten resistant (alive) mosquitoes generated by the SureSelect experiment. The pF_st_ (https://github.com/vcflib/vcflib, accessed on 6 April 2021) tool was run on the freebayes-predicted variants (applying flag “—type GL” in pF_st_). The analysis was restricted to bi-allelic coding sequence variants, and the resulting *p*-values were false discovery rate adjusted using qvalues [52] package in R, and with a threshold of 0.05 applied.

### 2.5. Gene Expression of APL1, TEP and LRIM Genes

The RNASeq data for 46 replicates of *An. funestus* (first published in Weedall et al., 2019) from Ghana, Uganda, Cameroon, Malawi, FANG and FUMOZ were aligned to the genome using the subread aligner (2.0.1) and quantified using featureCounts of the Subread package [53]. The raw counts were converted to transcripts per million (TPM) values (following http://ny-shao.name/2016/11/18/a-short-script-to-calculate-rpkm-and-tpm-from-featurecounts-output.html, accessed on 17 September 2021) to allow comparison between the genes, and the average TPM per gene calculated from all of the replicates combined. These RNASeq data were generated from the total RNA of mosquitoes that survived exposure to pyrethroids (resistant) and DDT, or that were not exposed to insecticides (control) and the resistant (FUMOZ) and susceptible (FANG) laboratory strains. A differential gene expression analysis between countries (combined by insecticide treatment) and the FANG and FUMOZ laboratory strains was performed on the raw counts in DESeq2, using the iDEP server (version 94) [54]. The counts were filtered to include only genes with a minimum count per million of 0.5 in at least four libraries. All of the 15 possible pairwise contrasts were tested, with a false discovery rate cut-off of 0.05 used to accept significance. Of the 13,144 genes that passed iDEP filtering, only the results for immunity genes of the AMP, *APL*, *LRIM*, *TEP* family genes and their paralogues were investigated further.

### 2.6. Identifying Shared Polymorphisms and Recombination between APL1 Paralogues

The discordantly mapped read pairs, in which each read of the pair mapped to a different paralogue, were quantified for each population. These discordantly mapped pairs may result from gene conversion homogenizing sequences between paralogues, or, if spurious, lead to inflated diversity estimates of *APL1* and paralogues. The discordant mappings were removed for each paralogue by filtering the unmapped, secondary and supplementary alignments in Samtools. The variants were recalled for these genes in Varscan, annotated with SnpEff, and gene-wide diversity metrics and coverages re-inferred with SNPGenie and bamstats05, respectively. To identify the shared polymorphisms in the stringently remapped data, a codon-aware alignment of the five paralogues was created in macse2 (v2.05), and the Varscan-predicted synonymous and non-synonymous SNPs per gene were converted to positions in the multiple sequence alignment. The VCF files for each position were then combined to identify the shared polymorphisms’ positions. Secondly, the locations of discordant mappings with predicted insert sizes greater than 10,000 bp were identified and counted for each paralogue by intersecting the read pair mapping with gene annotations in bedtools for PoolSeq and SureSelect data. A Venn diagram of the shared polymorphisms between *APL* paralogues was created, using jvenn [55].

The recombination between the paralogues was tested with GARD (Genetic Algorithm in Recombination Detection) on the Datamonkey Adaptive Evolution Server. GARD was run with defaults and with ‘General Discrete’ site-to-site variation and three rate classes, to check consistency between parameters. The coiled coil domains were predicted for each paralogue, using DeepCoil2 [56].

## 3. Results

### 3.1. Immune Gene Complements of An. funestus

We identified one copy of *APL1* (*AFUN018743*) syntenic with *APL1C* in *An. gambiae* and 4 paralogues, 13 *LRIM* genes and 36 *TEP* genes in the *An. funestus* genome annotation. Their designations and chromosomal locations are given in Appendix A. The *An. funestus APL1* and the four paralogues all encode a leucine-rich repeat domain and two 3′ end coiled coils, similar to their orthologues in *An. gambiae*. None of the *An. funestus APL1* genes, however, share the repeated amino acid motif ‘P-A-N-G-G-L’ present in the 5′ end of *An. gambiae APL1C*. None of the four paralogues of *APL1*: *AFUN018581*; *AFUN000279*; *AFUN000288* and *AFUN000597* are syntenic with *APL1A/B/C* of *An. gambiae*. All five of the *An. funestus* genes are orthologous to the *APL1* of *An. stephensi* (*ASTEI02571*) which lacks paralogues [20], while only *APL1*/*AFUN018743* is syntenic. The phylogenetically closest species to *An. funestus* with VectorBase genome annotations are *Anopheles culicafacies* and *Anopheles minimus*. Both species encode three genes orthologous only to the five *APL1*-like genes of *An. funestus*. The *TEP* and *LRIM* genes, *TEP1* (*AFUN018758*) and *LRIM1* (*AFUN005964*), orthologues were defined by synteny to their *An. gambiae* equivalents in VectorBase (*AGAP010815* and *AGAP006348*, respectively). Most of the *APL1*, *LRIM* and *TEP*-like genes are present on chromosome 3, with eight on chromosome 2 and zero on the X chromosome.

Two anti-microbial peptide-family gambicin genes (*AFUN006610* and *AFUN006611*) are encoded in the *An. funestus* genome, which occur consecutively on chromosome 2 (around position 65.13 mb). A third consecutive gene, *AFUN006612*, is a VectorBase paralogue of *AFUN006611* but is annotated as an unspecified product. Other AMP genes identified include the four cecropins, four defensins and one attacin gene; only diptericin is lacking from the *An. funestus* genome.

### 3.2. APL1 Has Elevated Diversity

The complete PoolSeq and SureSelect datasets consisted of 12,968 and 952 genes with SNPgenie polymorphism data, respectively. In the PoolSeq data, 229 genes had a π_N/_π_S_ ratio greater than one, 36 of which were annotated with functions, including one of the gambicin genes (*AFUN006610*), two of the defensin genes (*AFUN016516* and *AFUN016588*), two salivary gland proteins (*AFUN016070* and *AFUN016250*) and a cuticular protein RR-1 (*AFUN000936*). Of the genes included in the targeted-enrichment data, π_N/_π_S_ was greater than one for the same gambicin, but not for the salivary gland and cuticular protein genes.

In both PoolSeq and SureSelect data, *An. funestus APL1* is an outlier in diversity levels compared to genes of a similar length (nonsynonymous sites polymorphism π_N_, Figure 1) with a π_N_ of 0.027 and a π_S_ of 0.036 in the PoolSeq data (Table 1 and Appendix A). Both were elevated against genome-wide averages of 0.002 and 0.021 for π_N_ and π_S_, respectively (Appendix A). This was consistent across the populations, including the insecticide-susceptible lab strain FANG, with π_N_ ranging from 0.023–0.030 (Appendix A). The *APL1* paralogues also had elevated diversities, although to a lesser extent (Table 1). When ranked by PoolSeq π_N_ values, *APL1* had the 15th highest values of the 12,968 genes included, and only three of those with greater π_N_ were from genes with greater than 400 nonsynonymous sites (Appendix A). The immunity genes of the *LRIM* and *TEP* families do not show similarly elevated diversities, with the highest per gene π_N_ value of 0.013 (*AFUN005964*/*LRIM1*) and 0.017, respectively (Appendix A), nor do any of the genes from these families have a π_N_/π_s_ ratio of greater than one. The *TEP* gene with the highest π_N_ (*AFUN019003*) had no synteny to other *TEP* genes in VectorBase. The average coverage of each gene from the PoolSeq data indicated that *APL1* and paralogue *AFUN018581* have elevated coverage versus genome-wide means across the populations (Table 2), however, the coverage profiles of these genes were both lower (less than two-fold genome averages) and dissimilar in shape to those of known *An. funestus* duplication events (Appendix A, for comparison with a known duplication). Instead of a sharp well-defined region of doubled (or more) coverage, as seen for duplications, the regions of increased coverage are non-uniform across these genes. This suggests that an alternative explanation, such as gene conversion, explains the observed coverage variability.

### 3.3. Haplotype Analyses and F_st_ between Countries Reveals No Clustering by Resistance or Origin

The haplotype analyses of *APL1* (Figure 2) revealed a high degree of intermingling among the regions and resistance with no discernible groupings. Most of the haplotypes (145/160) were unique and did not group with any other sequences, the largest cluster was of six FANG sequences, followed by a cluster of three, also FANG, sequences. *TEP1* was very similar in its overall lack of shared haplotypes, with 144 of 160 being unique. Only two clusters of eight sequences each, both consisting entirely of FUMOZ sequences, were present. For both of the genes, the mutational separation between haplotypes was high (represented by circles on connecting nodes in Figure 2). *LRIM1*, by contrast, only presented two clusters across all of the 160 haplotypes, one of 152 sequences and one of 8, which were separated by only one mutation. Tajima’s D tests for detecting population structure or selection were non-significant for all three genes.

Pairwise F_st_ among the PoolSeq populations revealed a slightly lower average global F_st_ for the *APL1*, *TEP1*, *LRIM1* genes at 0.13, 0.12 and 0.15, respectively, versus a chromosome-wide estimate of 0.16 (Appendix A), while the two gambicins had F_st_s of 0.17 and 0.13. Only six of the genes had global F_st_s greater than 0.3 of the 62 immune genes tested. Four of these were *LRIM* genes, one was *TEP1* and one an *APL1* paralogue (AFUN000279), however the *APL1* paralogue only had two SNPs contributing to the F_st_ estimate, whereas the other genes had 33–143 SNPs contributing (Appendix A). Between the dead and alive SureSelect individuals, after filtering for biallelic variants, 146 positions were included in the F_st_ analysis of the *APL1* gene within populations. No positions were significantly different between the dead and alive individuals for each population tested.

### 3.4. Gambicin and Other Immune-Related Genes Are Also Highly Polymorphic

Two further immunity-related genes had elevated levels of diversity across PoolSeq populations and SureSelect data, a gambicin (*AFUN006611*) and a fibrinogen domain containing gene (*AFUN019026*) (Appendix A). The fibrinogen gene has six VectorBase paralogues, of which only one, *AFUN014704*, also had elevated diversity restricted to southern African populations (Appendix A). Gambicin (*AFUN006611*) also had a π_N_/π_s_ ratio greater than one in nine PoolSeq populations and a SureSelect π_N_/π_s_ of 1.55. This gene has two nearby gambicin paralogues in the *An. funestus* genome (*AFUN006610* and *AFUN006612*). The AFUN006610 also had an average PoolSeq π_N_/π_s_ greater than one at 1.37 while *AFUN006612* did not, at a π_N_/π_s_ ratio of 0.09 (Appendix A); *AFUN006610* was not included in the SureSelect baits hence no estimate was made from these data. The haplotype networks differed in topology between the two gambicin genes (Appendix A). The *AFUN006610* gene was dominated by one haplotype of 142/160 sequences, whereas for *AFUN006611* the largest cluster was of 59 sequences with smaller clusters of up to 12 sequences. Unlike for *APL1* and *TEP1*, the mutational distances between clusters were low, with at most two mutations separating the clusters in *AFUN006611* and only one in *AFUN006610*. Neither gene showed a separation between origin and resistance status of the included mosquitoes. The average coverages of gambicin genes were 39, 40 and 39-fold for *AFUN006610*, *AFUN006611* and *AFUN006612*, respectively, which did not indicate a possible duplication event underlying the elevated diversity. Other AMP genes lacked the consistency in high π_N_/π_s_ ratios of gambicins, with only a cecropin (*AFUN000369*) having elevated levels in the three Mozambique pools from the 2002, 2016 and FUMOZ resistance strains at 1.09, 0.98 and 0.98, respectively.

### 3.5. APL1 Expression Is Greater Than Its Paralogues and a Subset of AMPs Are Very Highly Expressed

Of the five *APL1* paralogues, *APL1* is expressed more highly across Africa than the other genes, at an average expression level of 102.43 (TPM, Appendix A). The next closest *APL1* paralogue in expression is *AFUN000597*, at a mean TPM expression of 17.45. It is important to note, however, that there may have been some artefactual expression between the *APL1* locus and paralogues in hard-to-discern directions, as the underlying gene conversion will have affected the RNASeq read mapping. Indeed, cross-mapping reads were used to support the inference of gene conversion as part of this study. One *TEP* gene (*AFUN02066*), syntenic with *TEP15* and *TEP2* in *An. gambiae*, had double the average of *APL1* at 201.30 TPM. Two of the *LRIM* genes were also of higher average expression than *APL1* (*AFUN003917* and *AFUN005964*), while the *TEP1* gene *AFUN018758* was slightly lower at 73.54 TPM (Appendix A; per-replicate and average normalized expressions of *APL*/*TEP* and *LRIM* genes). For the AMP genes, three cecropins (*AFUN011465*, *AFUN015822* and *AFUN015823*) were expressed at by far the highest TPM of all of immune genes tested (means 1448, 2691, 2031 TPM, Appendix A) followed by a defensin (*AFUN006915*) at 795 TPM. The two gambicin genes and their paralogue were expressed at much lower TPMs of 14 (*AFUN006610*), 1 (*AFUN006611*) and 1 (*AFUN006612*) TPM, respectively.

The RNASeq results for the *APL1* revealed three contrasts between FANG, Ghana and Uganda, all of them versus Cameroon were significant, with expression being higher in Cameroon in each case (Appendix A). For *TEP1*, six contrasts were significant, in which Ghana and Uganda both have lower expression than Cameroon, FANG and FUMOZ. The *LRIM1* locus was significantly lower expressed in Ghana versus both Cameroon and FANG. For all three categories of immune gene, log_2_-fold changes were not large (<2). For the two gambicins and the paralogue, only *AFUN006610* was significant across ten of the fifteen contrasts, and expression was highest in susceptible FANG and low in resistant FUMOZ (Appendix A). Of the cecropin and defensin genes, *AFUN011465* was the most interesting. It was significant for six contrasts, including all five contrasts involving FANG, where it was expressed at an average of 2340 TPM across the FANG replicates, versus 1244 for all of the other replicates combined (Appendix A).

### 3.6. Discordant Read Mappings Consistent with Gene Conversion between APL1 Paralogues

We isolated the discordant read-pair mappings in the PoolSeq and SureSelect datasets for *APL1* and its paralogues from the combined Africa-wide datasets for each. This revealed that discordant mappings of paired reads occur extensively between paralogues (Figure 3 and Appendix A). For all five paralogues, discordant mappings were identified across the two exons of each gene. These mappings were biased to exon 2 for each paralogue in PoolSeq and SureSelect data, which form most of the coding sequence across the *APL* paralogues. In *AFUN000288*, no discordant mappings were observed across exon 1 (Appendix A). The only notable gene which experienced cross-mapping of reads to the *APL1* paralogues was gene *AFUN020339*. However, this was an artifact of paralogue *AFUN000279*, being encoded in an intron of this gene. The *APL1*/*AFUN018743* locus shared most discordant read-mappings with *AFUN015851*, the paralogue which also had high average PoolSeq coverage.

By converting the variant positions between the *APL1* paralogues to those of the multiple sequence alignment and verifying that the same reference/ancestral and alternative allele occurred at overlapping positions, we identified shared polymorphisms. The shared polymorphisms remained after discordant read mappings were removed from the variant predictions (Venn diagram, Figure 4). The greatest overlaps were between *APL1*, *AFUN018581* and *AFUN000279* at the 25 and 20 variant positions, respectively. Both GARD models predicted 20 breakpoints across the *APL1* paralogue alignment, and each was significant for breakpoints versus rate heterogeneity across the alignment. The longest unbroken segment of the alignment was between positions 257–1320 of the 3075 bp long alignment. This corresponded to the coding region present only in *AFUN000288* but was equivalent to a length of 72 bp only in the other paralogues, a length that was consistent with the other breakpoint lengths predicted by GARD (Appendix A).

Two coiled-coil domains were predicted by DeepCoil in the 3′ region of all five paralogues (Appendix A). In *APL1*, the non-synonymous mutations occurred more frequently in these coiled-coil domains in PoolSeq data: 56 such sites were identified in the 313 bp covering these two domains, versus 188 across the 1593 bp of remaining coding sequence. Furthermore, the coiled-coil domains contained 10 multi-allelic nonsynonymous SNPs, a rate twice as high (0.032 per non-synonymous site vs. 0.016) as for the rest of the gene, which has 25 such multi-allelic nonsynonymous variants. This coiled-coil region was not the region with the most discordant mappings in *APL1* (Figure 3).

## 4. Discussion

The *An. funestus APL1* gene and its paralogues encode a 5′ signal peptide, LRIM domain and 3′ end coiled-coil region. This is in line with *An. stephensi APL1*, to which *An. funestus* is more closely related than *An. gambiae s.l.* [20,57]. None of the *An. funestus APL1* genes contain the variable ‘PANGGL’ motif present on the 5′ end of *An. gambiae APL1C* and *APL1A* [18], nor does the *APL1* gene of the more closely related *An. stephensi* [20]. None of the genes investigated here, including AMPs, had patterns of diversity restricted to their regions or resistance status (Figure 2 and Appendix A), although FANG and FUMOZ formed clusters in *APL1* and *TEP1* haplotype networks, respectively, whereas almost zero field samples had concordant haplotypes.

### 4.1. Gene Conversion Explains Elevated APL1 Diversity in An. funestus

The three genes that form the pathogen-eliminating complement system in *Anopheles* have a distinct pattern of diversity to their *An. gambiae* counterparts. The elevated diversity *APL1* is similar to that of *An. gambiae,* but no population exhibits the hallmarks of a selective sweep in the same way as *APL1C Plasmodium* resistance alleles in *An. coluzzii* [18]. Unlike *An. gambiae*, the *TEP* gene diversities were not outliers compared to background levels [19], although the highest π_N_ values were elevated (>0.015) for four of these genes (Appendix A). The *LRIM* genes exhibited the least elevated diversity of the three types, similar to the *LRIM* genes in *An. gambiae* [19,24]. The results were consistent across PoolSeq and SureSelect. No genes from across each category appeared to have undergone a recent selection sweep, which would have been shown by reduced π_N_ and π_s_ at the affected locus and surroundings. We also found no link between insecticide resistance and allele frequencies for the principal *APL1* locus (*AFUN0018743*). Thus, in these data, a link between the insect complement system and vector competence cannot be made. By contrast, the increased *GST* expression in *An. funestus* does lead to a higher *Plasmodium* load through the metabolism of parasite-killing ROS and/or a fitness trade-off that decreases the robustness of immune responses [32].

This observed diversity is not an artifact of gene duplication of the *APL1* gene or reads mapping between paralogues due to high sequence similarity. When read-pairs that mapped between *APL1* and its paralogues were removed, the genetic diversity of *APL1* (π_N_ and π_S_) remained high in both the pooled and target-enriched datasets. Furthermore, the shared polymorphisms between paralogues were identified after the removal of discordant read-pairs filtered from the datasets. This is consistent with gene conversion between the paralogues of *APL1* that maintains high diversity at the principal locus. The paralogue alignment-based GARD approach detected many recombination events among the five *APL1* paralogues, which is concordant with gene conversion [58]. The *An. stephensi APL1* diversity sampled from Iran did not exhibit the elevated polymorphism of *An. funestus* found in all populations sampled here [20]. In *An. gambiae,* the elevated *APL1* diversity was not associated with an increased interspecific divergence through relaxed constraint or a higher-than-average mutation rate [18]. In fact, lower inter-specific divergence was observed, consistent with adaptive maintenance of variation.

### 4.2. Gene Conversion Acts on APL1 Orthologues in Other Anopheles Species

The observed gene conversion-mediated increased diversity in *An. funestus APL1* mirrors that found for the *TEP1* gene in *An. gambiae* [19]. It is also likely true for the *APL1* paralogues in *An. gambiae,* as the polymorphisms and the PANGGL motif are shared between loci [18]. The *An. gambiae s. l. TEP1* was shown to exchange diversity with *TEP5* and *TEP6* through gene conversion [19]. However, the high diversity in *TEP1* was principally due to the presence of two divergent allele classes, one of which provides resistance to *Plasmodium* infection. When analyzed separately, the two allele classes exhibited diversity levels close to other *An. gambiae* loci [19]. This is similar to the *An. funestus* orthologue of *TEP1*, and other *TEP* genes, which do not show elevated polymorphisms across Africa (Appendix A), suggesting no such divergence of haplotypes in this species. However, the granularity of the *TEP1* haplotype network (Figure 2b) suggests that processes may be occurring in this gene that are too subtle to detect by the population genetic metrics applied here.

Gene conversion can lead to increased diversification at a locus under a model in which multiple donor genes contribute diversity, either reciprocally or to a recipient single locus [59]. This is not restricted to complete genes, as the pseudogenes of the spirochaete *Borrelia hermsii* and African Trypanosomes can also donate diversity to surface protein genes, in order to evade host immune responses [60,61]. Here, we found that the *APL1 An. funestus* locus exhibits high diversity, as, to a lesser extent, do its four paralogues. This is a different outcome to the more frequently observed gene conversion between a single donor and recipient locus, which leads to sequence homogenization, which has occurred between the tandemly duplicated insecticide-resistance-conferring *Cyp6P9a* and *Cyp6P9b* loci in *An. funestus* from Benin [36]. Kurosawa and Ohta (2011) proposed that a highly expressed gene could become the principal recipient locus among a set of paralogues experiencing reciprocal gene conversion [59]. We observed such a gene expression relationship between *APL1*, which is both the most highly expressed and genetically diverse gene among the five paralogues. This relationship is hypothesized to result from the regulation of gene expression by local DNA accessibility, which is controlled by epigenetic modifications on surrounding chromatin. A gene more highly expressed than others in its gene conversion network will have more accessible DNA, which enhances the probability of homologous recombination occurring at this locus [59].

The elevated diversity of *APL1* was maintained in *Anopheles gambiae sensu. stricto.* post-duplication of *APL1* into three copies [18]. This implies that elevated diversity is an important component of *APL1* function, as a simple model of sub-functionalization of gene copies post-duplication, through differential pathogen targeting, for example, would predict a reduction in allelic diversity at each locus.

### 4.3. Elevated π_N_ in the An. funestus Anti-Microbial Protein Gambicin

The high gambicin gene π_N_/π_s_ ratios indicate positive selection, alternatively, balancing selection could explain both of the π_N_/π_s_ greater than one and the outright high levels of π_N_ shown in Figure 1 [62]. Alternatively, this locus could be undergoing gene conversion, as is likely occurring at *APL1*. One test for balancing selection would involve testing for trans-species polymorphisms at these two loci in other *funestus* group species. Such trans-species polymorphisms are commonly found in AMPs among the *Drosophila* species [62]. We also note that this is a relatively short gene with ~61 synonymous sites, which may make the π_N_/π_s_ ratio more susceptible to stochasticity than longer genes. Interestingly, gambicin is the only AMP among Dipterans that has been previously associated with positive selection [23,33,63]. Among the wider insects, evidence for positive selection on AMPs is patchy, despite the natural expectation that the AMPs will evolve rapidly under selection from microbes [33]. This has been explained by the diversity of the AMPs that insects encode, creating a dynamic environment which prevents selection on any one component [64]. Rather, selection is hypothesized to act on the speed and efficiency of transcription and translation of AMPs post-infection [33,65].

Current models of insect cecropin AMP activity assume low levels of expression in the midguts, fat bodies or reproductive tracts, until an immune challenge ramps up transcription [66]. Our observation of three cecropins and a defensin expressed at much higher levels (>700-fold plus) than the other AMPs indicates that this is an incomplete model. Especially so, as expression of three of these four genes was consistent across African populations and laboratory strains, which suggests constitutive high expression of these genes in the face of diverse environmental pathogen challenges. The fourth gene, *AFUN011465*, a cecropin, is particularly interesting, due to both its high expression and greater expression in the susceptible FANG strain. We hypothesize that the FANG strain mosquitoes can invest more resources into expression of this immune gene, as they do not incur the energetic costs of metabolic resistance mechanisms versus the other strains. The fitness costs of metabolic resistance in *An. funestus* are well-established with respect to life-history traits [67,68]. Indeed, the opposite has been observed in the moth *Trichoplusia ni*, in which investment in immunity has its own negative effect on developmental fitness [69]. Conversely, the AMP genes may be ‘cheap’ to express constitutively and therefore have little to no effect on overall fitness, as found for diptericin in *Drosophila melanogaster* [70]. Under this scenario, we would not expect a trade-off to explain the higher *AFUN011465* expression in susceptible FANG, although the added burden of an additional factor in resistance-conferring gene expression may make AMP expression more costly than otherwise. Linking fitness traits measured phenotypically to immune homeostasis through gene expression would further our understanding of the mechanisms underlying resistance trade-offs. This hypothesis can be tested by pathogen survival analyses on different strains of *An. funestus* versus FANG in controlled conditions, without insecticide exposure. Such an experiment would require some knowledge of the pathogen range of the *AFUN011465* cecropin protein with respect to *Plasmodium*, bacteria and other threats.

## 5. Conclusions

*An. funestus* has high diversity at the key immune complement factor, *APL1*, which is most likely due to non-homologous gene conversion between the five paralogues of this gene. This is in line with the *APL1* genes in *An. gambiae s. l.* in sub-Saharan Africa, but not the more closely related *An. stephensi*. The existence of the non-syntenic APL1 paralogues may provide neo-functional diversity that liberates *APL1* in *An. funestus* from the constraints observed in *An. stephensi,* which encodes a single copy of *APL1*. In future experiments, the mortality effect of silencing *APL1* through RNAi (see [71] for similar) in response to pathogen challenge would help establish sub-functionalization at this locus. The combinatorial silencing of paralogues would also help define which paralogues act functionally or as wells of genetic diversity alone. We hypothesize that open chromatin at the most expressed paralogue (*APL1*/*AFUN018743*) may explain preferential replenishing of this locus with diversity from other paralogues. The *An. funestus TEP1* gene that *APL1* interacts with does not exhibit high diversity or the selective sweep signals seen in *An. gambiae s. l.*, nor do numerous paralogues of this gene. The final gene of the pathogen-eradicating complex, *LRIM1*, also lacks a pattern of diversity different to the background levels that are similar to other *Anopheles* species investigated to date. We also show that gambicin anti-microbial peptide genes have elevated nonsynonymous diversity. The gambicins are the only AMP genes that have been previously associated with positive selection in insects, and gene conversion or balancing selection could explain these observations. The cecropin and defensin AMPs were constitutively highly expressed across *An. funestus,* at odds with expectation and therefore warranting further investigation.

## Figures and Tables

**Figure 1 genes-13-01102-f001:**
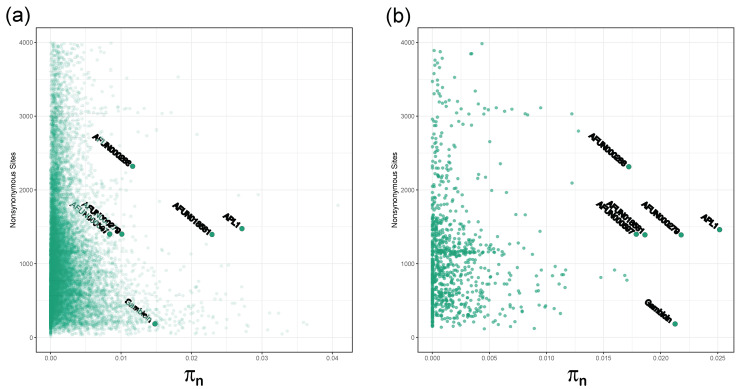
***Anopheles funestus APL1* (*AFUN018743*) is an outlier in non-synonymous diversity**. Number of non-synonymous sites versus π_N_ for all genes (**a**). PoolSeq data and a selection of potentially resistance associated loci; (**b**). SureSelect data. *APL1* paralogues also have high π_N_ relative to other genes, but to a lesser extent and with more variability between data types. The anti-microbial gene gambicin (*AFUN006611*) is also labelled.

**Figure 2 genes-13-01102-f002:**
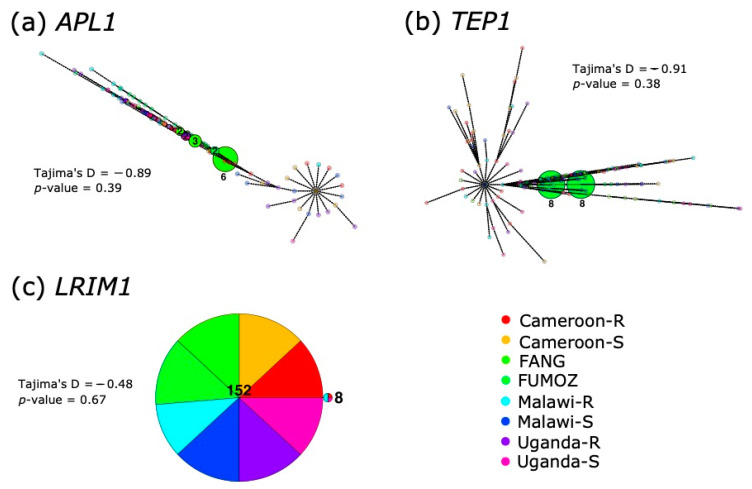
**Haplotype networks of (a) *APL1*; (b) *TEP1* and (c) *LRIM1* from SureSelect data**. Haplotypes are colored by origin (country or laboratory strain) and resistance (R) or susceptibility (S) of individual mosquitoes. Black circles on nodes indicate the mutational distance between haplotypes. Unique haplotypes were made translucent due to the degree of overlap along nodes. Tajima’s D and associated *p*-value are given for each gene adjacent to the haplotype network.

**Figure 3 genes-13-01102-f003:**
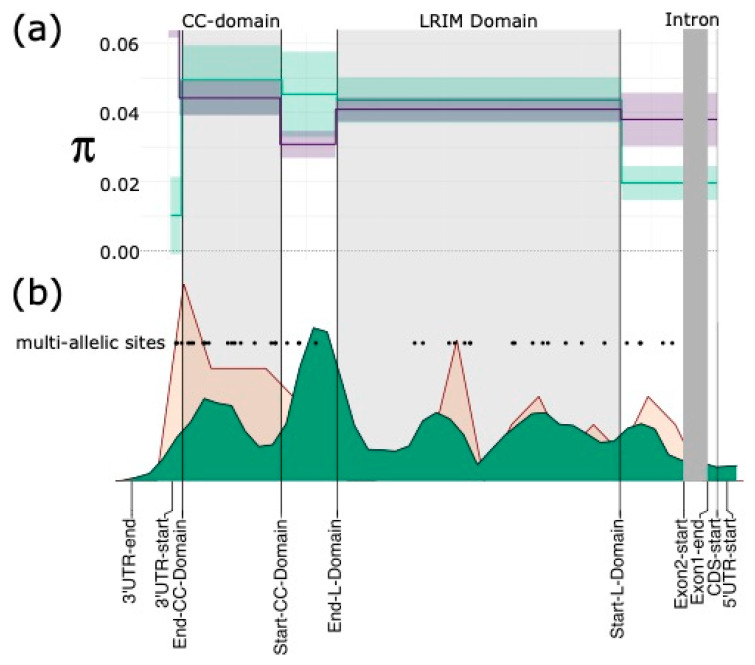
**Diversity of *APL1* domains and discordant read pair mappings.** (**a**) Nucleotide diversity (π) for non-synonymous sites in purple and synonymous sites in green; line is mean across twelve PoolSeq populations and shaded area is standard deviation; (**b**) Discordant read mapping density across the *APL1* gene-body is given in green in lower figure, density of multi-allelic positions is shown in salmon pink and positions of such sites are plotted as black dots labelled “multi-allelic sites”. CC-Domain is the region encoding two coiled-coil motifs and LRIM domain, both are shaded in light grey. The single intron is shaded in dark grey. Domains, exon and untranslated region positions are given on the *X*-axis.

**Figure 4 genes-13-01102-f004:**
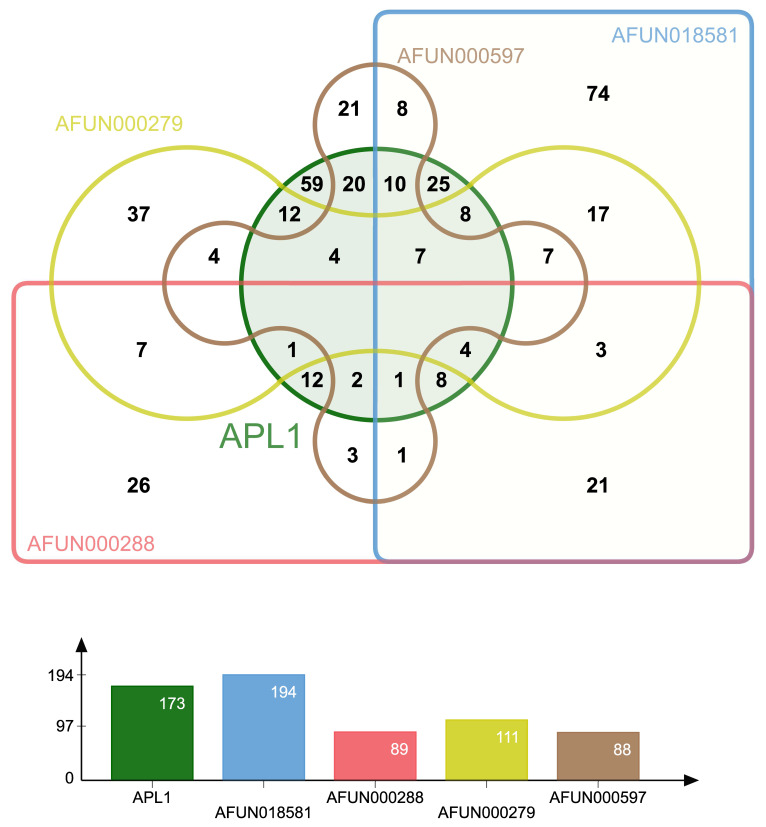
**Venn diagram and bar chart of shared polymorphisms for APL1 paralogues.** Polymorphisms that occur in the same alignment position of the same reference and alternate alleles between paralogues after removal of discordant read mapping. Each gene is labelled with a different color, total number of shared positions per gene is given in the bar chart below.

**Table 1 genes-13-01102-t001:** Diversity estimates for *APL1* and paralogues in *An. funestus*.

Gene	Nonsynonymous Sites	Synonymous Sites	π_N_	π_S_
PoolSeq				
*APL1/AFUN018743*	1475.80	426.20	0.027	0.036
*AFUN018581*	1396.35	409.62	0.023	0.036
*AFUN000279*	1399.86	406.14	0.010	0.012
*AFUN000288*	2320.70	688.30	0.012	0.019
*AFUN000597*	1401.36	404.64	0.008	0.008
**SureSelect**				
*APL1/AFUN018743*	1461.21	418.62	0.025	0.027
*AFUN018581*	1389.97	402.91	0.019	0.023
*AFUN000279*	1389.10	402.13	0.022	0.028
*AFUN000288*	2313.82	689.23	0.017	0.025
*AFUN000597*	1398.40	404.38	0.018	0.020

Nonsynonymous sites = number of nonsynonymous positions across gene; Synonymous sites = number of synonymous positions across gene; π_N_ = nucleotide diversity at nonsynonymous sites; π_S_ = nucleotide diversity at synonymous sites.

**Table 2 genes-13-01102-t002:** Average gene coverages from combined PoolSeq data versus background genome-wide averages for *APL1* and its paralogues.

Gene	Original Coverage	Discordant Read Filtered Coverage
*AFUN018743*	48.31	42.79
*AFUN018581*	49.55	43.16
*AFUN000279*	20.79	17.06
*AFUN000288*	23.12	20.48
*AFUN000597*	14.60	12.15
All genes	33.98	N/A

## Data Availability

Read data for PoolSeq, SureSelect and RNASeq analyses is available in the European Nucleotide Archive under accessions PRJEB13485, PRJEB24384, PRJEB35040, PRJEB24351, PRJEB24520, PRJEB47287, PRJEB48958 and PRJEB24506.

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
