# Peer review of "Gene Conversion Explains Elevated Diversity in the Immunity Modulating APL1 Gene of the Malaria Vector Anopheles funestus"

_genes, 2022, doi:10.3390/genes13061102_

Round 1

Reviewer 1 Report

Gene conversion explains elevated diversity in the immunity modulating APL1 gene of the malaria vector Anopheles funestus
Hearn et al

APL1 encodes a leucine rich repeat protein involved in Anopheles immunity. APL1 paralogs were previously described as exceptionally polymorphic in the A. gambiae/ coluzzii genome. The ancestral state of APL1 is a single gene in all Anopheles except an African clade including the A. gambiae complex. The ancestral APL1 is functionally distinct from the diverged family paralogs, but the population genetics was examined only in a study of A. stephensi (Aste APL1) from Iran, which indicated that the ancestral single APL1 gene was much less diverse than the derived copies.

Here, the ancestral single APL1 gene is genetically characterized in pooled sequence reads and bait-selected enrichment from individual mosquitoes of A. funestus. The pool-seq method is not ideal as compared to individual whole genome sequencing, but is a reasonable cost-benefit compromise. The analyses taken together are justified by the authors and the results are coherent across the methods, and thus there are no particular technical concern. The report is of high quality and is appropriate for publication as is, with minor textual revision as mentioned. The writing is clear and the work is thoughtful, containing a lot of interesting ideas to follow up.

Interestingly, in addition to the Afun APL1 gene, which is orthologous and syntenic to Aste APL1, four other Afun paralogs are identified that are also orthologous but not syntenic to Aste APL1. This finding indicates that there could be biological differences among carriers of the “ancestral single copy APL1 gene”, exemplified so far with a detailed study only in A. stephensi, which does not have APL1 paralogs. Thus, the single APL1 copy may turn out to be not functionally uniform among Anopheles, and the existence of non-syntenic APL1 paralogs in at least A. funestus may provide neofunctional diversity that could liberate the single APL1 gene from the essentiality as observed in A. stephensi. This would require further experiments, for example to test the mortality effect of silencing Afun APL1. These kinds of experiments are beyond the scope of the current report and no further work is required here. However, this could be mentioned in the Discussion, because the current findings deepen the biological understanding of APL1 and relatives.

The claim of elevated genetic diversity of Afun APL1 is convincing. It is consistent with reports for the expanded APL1 family in the A. gambiae complex, but not with A. stephensi, carrying only one APL1 copy. The high diversity of Afun APL1 is another indication that Afun APL1, perhaps because of neofunctional cooperation by its genomic paralogs, may not display the essentiality of Aste APL1. The presented data are consistent with a mechanism of gene conversion among Afun paralogs, and the suggestion that higher expression of Afun APL1 than its paralogs could lead to an open chromatin conformation that would promote gene conversion from donor paralogs is plausible.

Specific points:
Line 64 typo of Anopheles christyi
Line 247 “orthologous to APL1B of An. stephensi (ASTEI02571)” This should read APL1, not APL1B
line 466 Sentence fragment "We hypothesize that"

Reviewer 2 Report

1. Line 25: "hypothesise" change it to "hypothesised"

Author Response

  1. Line 25: "hypothesise" change it to "hypothesised"

Response: This has not been changed as the hypothesis is a conclusion for further work. I have added “In conclusion, we hypothesise” to the start of this sentence at line 25 to make this clearer.